# Tailoring dengue health communication: Survey-based strategies to reduce message fatigue across risk areas

Chia-Hsien Lin[1]*, Yen-Jung Chang[2], Hung-Yi Lu[3,4]

1 Independent Researcher, New Taipei City, Taiwan, 2 Department of Health Promotion and Health Education, National Taiwan Normal University, Taipei City, Taiwan, 3 Department of Communication and Graduate Institute of Telecommunications, College of Social Sciences, National Chung Cheng University, Chiayi County, Taiwan, 4 School of Medicine, College of Medicine, National Sun Yat-Sen University, Kaohsiung City, Taiwan

* chlin1983@gmail.com

## Abstract

### Background

Dengue remains a significant public health challenge in Taiwan, particularly in the southern region where *Aedes aegypti* and *Aedes albopictus* mosquitoes thrive. Despite nationwide dengue prevention campaigns, message fatigue—where individuals become disengaged due to repetitive messaging—may reduce the effectiveness of public health communication. This study analyses trends and associations between key predictors—such as age, sex, education, job, and perceived risks (optimistic bias, perceived prevalence, and perceived severity)—and message fatigue across different dengue-risk areas in Taiwan.

### Methodology/principal findings

A cross-sectional online survey was conducted from October 25 to November 13, 2023, among 814 adults across Taiwan. Participants were classified into high-risk (southern Taiwan) and low-risk (other regions) groups. Message fatigue was assessed using the message environment (ME) and audience response (AR) dimensions. Perceived risks (perceived prevalence, severity, and optimistic bias) and demographic variables (sex, age, education, job experience, and residency) were analyzed using Poisson and Negative Binomial regression models.

### Findings

The findings revealed that message fatigue varied by region and demographic factors. In high-risk areas, individuals with lower education levels exhibited higher AR fatigue (boredom and disengagement) (p = 0.04). In low-risk areas, males (p = 0.02), individuals with lower education (p = 0.01), and those with optimistic bias (p = 0.02)

**Data availability statement:** All relevant data are within the manuscript and its Supporting Information files.

**Funding:** The author(s) received no specific funding for this work.

**Competing interests:** The authors have declared that no competing interests exist.

reported significantly higher ME fatigue (perceived overexposure). Despite uniform nationwide messaging, participants in high-risk regions exhibited greater message fatigue, possibly due to habituation.

## Conclusions/significance

Message fatigue is associated with regional dengue risk, demographic factors, and perceived risk. A uniform health communication approach may not be effective across different populations. In high-risk regions, message fatigue is more pronounced. Repetitive warnings may contribute to disengagement, suggesting a need for communication strategies that reduce redundancy and emphasize localized, actionable information. In low-risk areas, messages should focus on engagement, particularly for men, who exhibited higher fatigue in this study. Simplified messaging can reduce cognitive overload for lower-educated populations. Addressing message fatigue can enhance the effectiveness of dengue prevention campaigns and sustain public engagement in long-term health communication efforts.

### Author summary

Dengue continues to pose a major public health challenge in Taiwan, where the government promotes prevention through nationwide messaging. However, repeated exposure to these messages can lead to message fatigue, resulting in diminished attention to prevention efforts. This study analyzes trends and associations between demographic factors (age, sex, education, and job) and perceived risks (optimistic bias, perceived prevalence, and perceived severity) in relation to message fatigue in high- and low-risk dengue regions. We surveyed 814 adults across Taiwan and analyzed message fatigue in two dimensions: feeling overwhelmed by excessive messaging and feeling exhausted by repeated information. In high-risk areas, individuals with lower education levels were more likely to feel exhausted by dengue messaging, indicating that current strategies may not effectively address their needs. In low-risk areas, males, individuals with lower education levels, and those with an optimistic perception of risk were more likely to find dengue messages excessive and redundant, suggesting that existing communication efforts may not resonate with these groups. These findings indicate that dengue prevention messaging should be tailored to different audiences in varying risk areas. High-risk areas may benefit from more localized and specific guidance, while low-risk areas may require adjusted messaging strategies, such as more engaging formats or personalized communication, to better align with their target audiences. Adapting health communication to reduce fatigue could help sustain awareness and encourage preventive behaviors.

## Introduction

Dengue is a significant communicable disease globally, and Taiwan is no exception. The entire Taiwan is susceptible to dengue due to the presence of two major mosquito vectors, *Aedes aegypti* (*Ae. aegypti*) and *Aedes albopictus* (*Ae. albopictus*) [1]. However, dengue incidence, prevalence, and vector distribution vary across regions, which formed the basis for classifying risk levels in this study. The southern region, specifically Tainan, Kaohsiung, and Pingtung, accounted for over 98.6% of indigenous dengue cases between 1998 and 2023 [1]. Studies have shown significantly higher prevalence rates in the southern region compared to other areas, including the northern, central, and outlying island regions [2,3]. For instance, a study by Hsu et al. reported that between 2010 and 2015, the highest prevalence rate was 616 per 100,000 population in the southern region, compared to 47, 6, 4, and 4 per 100,000 population in the northern, central, eastern, and outlying island regions, respectively [2]. Another study indicated that dengue Immunoglobulin G seropositivity in Tainan was significantly higher than in Taipei, located in the northern region, based on a survey conducted in 2010 [3]. Regarding vectors, *Ae. albopictus* exists throughout Taiwan, while *Ae. aegypti* is present primarily in the southern region and parts of the eastern region [4–8]. Consequently, the southern region of Taiwan has historically experienced the highest dengue incidence, prevalence, and vector density, with both *Aedes aegypti* and *Aedes albopictus* present, categorizing it as a high-risk area. In contrast, the northern, central, and outlying island regions have consistently reported fewer cases, with only *Aedes albopictus* as the primary vector [4,9]. The eastern region presents a unique case where both mosquito species are present, but *Aedes aegypti* is restricted to a small area, which may limit its impact on transmission [4,8].

The current prevention and control strategies were established in 2003 [1]. These strategies are maintained year-round and consist of information, education, and communication (IEC) activities, as well as routine vector control. The Taiwan Centers for Disease Control (TW CDC) conducts IEC activities, which include health education campaigns through television advertisements, radio announcements, posters, the Internet, and social media platforms. These activities are uniform across Taiwan and aim to increase public awareness of self-protection measures, symptom recognition, and the management of water-holding containers on private properties. However, dengue continues to occur each year despite these year-round IEC activities, suggesting that these efforts may require modification to discover new approaches that motivate people to engage in prevention measures.

One reason that may reduce motivation to participate in disease prevention is message fatigue [10–12]. Message fatigue is a psychological state resulting from prolonged exposure to similar or repetitive messages, leading to negative reactions such as boredom, annoyance, and disengagement [13]. When individuals are overwhelmed with excessive messages on the same topic, they may experience mental exhaustion and boredom, resulting in message fatigue [14]. This state encompasses perceived overexposure, redundancy, exhaustion, and tedium [10,13,14]. Based on So et al.'s study, message fatigue consists of two dimensions: message environment (ME) and audience response (AR). The ME dimension describes a sense of being overwhelmed and frustrated by excessive and repetitive information about the message, while the AR dimension indicates boredom and annoyance with continuous and uninteresting messages [13].

Several demographic factors, such as age, sex, education, and job, have been identified as associated with message fatigue [15–18]. However, the patterns of these demographic factors are not consistent across studies. For example, Stockman et al. examined the level of fatigue toward HIV-prevention messages among three high-risk populations: men who have sex with men, heterosexuals, and injection drug users (IDUs). Their findings showed that younger IDUs experienced a higher level of HIV-prevention fatigue than older IDUs [15]. In contrast, Steiner et al. investigated whether the volume of automated text or interactive voice response messages influenced the likelihood of patients opting out of future messages, finding that seniors (aged 55 and older) experienced a higher level of message fatigue than younger individuals (aged 18–34) [19]. Regarding sex, a Polish study found that women reported significantly greater COVID-19 fatigue, particularly in terms of information fatigue [18], while men exhibited higher levels of anti-tobacco message fatigue in the United States [17].

Message fatigue has been observed in prevention messages for infectious diseases [11,14,15,20–22]. For instance, research by Guan et al. examined the impact of COVID-19 message fatigue on individuals' intentions to engage in preventive behaviors [11]. The results indicated that message fatigue leads to both reactance and inattention, which negatively affect individuals' intentions to engage in preventive behaviors, thereby mediating the relationship between message fatigue and these behavioral intentions [11]. In the context of HIV prevention messages, Stockman et al. investigated HIV-prevention fatigue among community-based samples from three high-risk populations in San Francisco. The study found that IDU participants exhibited a higher level of fatigue toward HIV-prevention messages compared to heterosexuals [15]. A report by UNICEF's Eastern and Southern Africa Regional Office revealed that a prolonged period of community awareness regarding Ebola prevention resulted in message fatigue [22]. However, few studies have explored message fatigue in the context of dengue.

Moreover, most studies on prevention messages for infectious diseases focus on the general population without considering varying risk levels or solely target high-risk populations concerning message fatigue. Few studies have evaluated message fatigue across groups with different risk levels, such as individuals residing in various areas but exposed to the same prolonged messaging. Groups with different risk levels may perceive disease risk differently, influenced by factors such as psychological and socio-economic conditions [23–25]. Additionally, perceptions of disease risks are associated with message fatigue [13,14]. Therefore, groups with different risk levels may experience varying degrees of fatigue.

Understanding message fatigue highlights the importance of tailoring communication to specific audience profiles, ensuring that future messages are both targeted and impactful. This survey study aimed to analyze trends and associations between key predictors—such as age, sex, education, job, and perceived risks (optimistic bias, perceived prevalence, and perceived severity)—and message fatigue across various dengue-risk areas in Taiwan. The findings from this study can inform global health communication strategies by identifying critical demographic and perception-based drivers of message fatigue. This knowledge can facilitate the development of more targeted and effective public health messaging across diverse regions with varying levels of risk.

## Materials and methods

### Ethics statement

This study utilized an online questionnaire that was reviewed and approved by the Research Ethics Committee (REC) of National Taiwan Normal University under approval number 202312HS019. Conducted anonymously, the study ensured that no personal identifying information was collected through the online platform, thereby safeguarding participants' privacy and confidentiality. Participants were informed about the purpose of the research at the beginning of the online questionnaire, and their voluntary participation was implied by their completion of the survey. Formal written or verbal consent was not obtained, as participation was anonymous, and no personally identifiable information was collected. While a lottery system was used to encourage participation, multiple responses were disqualified based on system detection rather than personal identifiers. The study adhered to ethical principles for research involving human participants, particularly regarding anonymity and the protection of data collected through online methods.

### Participants and procedure

The study was a cross-sectional study, which was part of a larger project involving an online survey on dengue perceptions and dengue-message fatigue. Participation was voluntary, and respondents were assured of confidentiality and anonymity. Participants were adults aged 18 and above residing in Taiwan, who reported their demographic information, perceived risks, and responses to questions assessing dengue-message fatigue. The questionnaire is in Chinese and consists of nine sections (S1 Table). To ensure a high completion rate, a lottery system was implemented. To prevent multiple participation, submitting multiple responses did not increase the chances of receiving an incentive and could lead

to disqualification.The questionnaire was distributed through the authors' social media networks, including Facebook and Line. A convenience sampling method was employed. The survey commenced on October 25, 2023, and concluded on November 13, 2023.

### Measurements of variables

**Message fatigue.** The online questionnaire was based on the measurement developed by So et al. according to the conceptual definition of message fatigue [13]. Similar to the study by So et al., the measurement of message fatigue included the ME and AR dimensions. Unlike the original study, this study's ME dimension consisted of six questions to represent overexposure and redundancy (Section VI 1–6 in S1 Table). The reduction from the original nine questions to six was due to the translation process from English to Chinese, during which some questions were combined or simplified without losing their original meaning. Additionally, one question was removed due to its impact on the Cronbach's alpha value. Consequently, five questions with a Cronbach's alpha of 0.85 were included in the analyses for this study (S2 Table). Thus, the number of questions was reduced while maintaining the consistency and clarity of the content. For the AR dimension, the number of questions was reduced from seven to five to represent exhaustion and tedium, due to translation-related adjustments similar to those made for the ME dimension (Section VI 7–11 in S1 Table). Of the five translated questions, one was removed during analysis because it negatively affected internal consistency. The remaining four questions yielded a Cronbach's alpha of 0.88 (S2 Table). All questions were measured on a 7-point Likert scale ranging from 1 (strongly disagree) to 7 (strongly agree), with response options including strongly disagree, disagree, slightly disagree, neutral, slightly agree, agree, and strongly agree [13].

Although ME and AR are Likert scale variables, which are ordinal data, this study treated the responses as count-like data (i.e., non-negative integers) for several reasons. First, this study focuses on analyzing trends and relationships between predictors and message fatigue without relying on assumptions about precise distances between ordinal categories. Second, while Likert scale data are not strictly count data, their discrete and bounded nature resembles count data. Third, when the underlying distribution of Likert scale data is skewed, raw scores tend to deviate from the true underlying distribution [26]. Therefore, treating Likert scale data as count data may better address this skewed data issue.

For each participant, ME and AR scores were calculated by summing their respective Likert-scale responses: five items for ME (perceived overexposure and redundancy) and four items for AR (exhaustion and disengagement). These summed scores served as the dependent variables in the regression models, allowing us to analyze their associations with demographic and risk perception factors (S3 Table). Higher ME scores indicate greater perceived overexposure to dengue messages, while higher AR scores reflect stronger exhaustion from repeated messaging.

**Perceived risks.** Perceived risks included perceived prevalence, perceived severity, and optimistic bias. Perceived prevalence was assessed with the question, "Do you think the current dengue outbreak in Taiwan is severe?" (Section I (1) in S1 Table), and perceived severity with, "Do you think the likelihood of death is high once infected with dengue?" Both were measured on 11-point Likert scales ranging from 0 (Not serious at all) to 10 (Extremely serious) (Section I (2) in S1 Table).

Optimistic bias can be assessed through direct or indirect comparisons [27]. This study employed an indirect comparison, in which respondents provided separate risk estimates for themselves and others using two questions: "How likely do you think it is that you will get a dengue infection?" (II (1) in S1 Table) and "How likely do you think it is that others around you will get a dengue infection?" (II (2) in S1 Table). These two questions were measured on 11-point Likert scales ranging from 0 (Very unlikely) to 10 (Very likely).

The optimistic bias score was calculated using indirect comparisons by subtracting the score of the first question (i.e., 'How likely do you think it is that you will get a dengue infection?') from the score of the second question (i.e., 'How likely do you think it is that others around you will get a dengue infection?') [27–29]. In this study, optimistic bias was categorized into three groups: realistic (score = 0) refers to respondents who believe they have an equal likelihood of getting a

dengue infection as others; optimistic (score > 0) refers to respondents who believe they are less likely to get a dengue infection compared to others; and pessimistic (score < 0) refers to respondents who believe they are more likely to get a dengue infection compared to others. The realistic group was set as the reference group for the optimistic bias variable. For analysis, the realistic group (score = 0) was designated as the reference category, as it represents a neutral state and provides a baseline for comparison with the optimistic and pessimistic groups.

**Demographic variables.** Demographic variables included sex, residency, education, job, and age. Sex was categorized as male and female. Residency was classified into five regions: north, central, south, east, and outlying islands. Education was classified into two levels: higher than high school and equal to or below high school. Job experience was categorized as having no job experience in dengue control and relevant job experience. Job relevance was established as a factor because issue relevance is associated with message fatigue. Lower relevance could amplify disengagement and reliance on heuristic judgments, leading to greater fatigue [30]. For each categorical variable, female, higher than high school, and no job experience in dengue control were set as the reference groups for sex, education, and job, respectively. Age was treated as a continuous variable.

**Risk classification.** For this study, regions were classified into two risk categories based on historical dengue case numbers and vector distribution. The southern region was defined as a high-risk area due to its higher incidence, prevalence, and the coexistence of both *Aedes aegypti* and *Aedes albopictus*. The remaining regions, including the northern, central, eastern, and outlying islands, were classified as relatively low-risk areas. Although *Aedes aegypti* is present in some parts of the eastern region, its distribution is limited, reducing its overall contribution to transmission risk.

## Statistical analysis

The independent variables included perceived risks (i.e., perceived prevalence, perceived severity, and optimistic bias) and demographic factors (i.e., sex, residency, education, job, and age). The overall distribution of message fatigue and perceived risks was presented using Likert plots, while demographic factors were described using descriptive statistics. To assess correlations among the independent variables before including them in the models, a Variance Inflation Factor (VIF) analysis was conducted for independent variables across different risk areas. Since independent variables include categorical variables, we used the Generalized VIF (GVIF) to assess multicollinearity. An adjusted GVIF (i.e., GVIF^(1/(2*Df))) value of less than two indicates no multicollinearity concerns [31].

The dimensions of ME and AR were assessed separately, as the questions described different feelings. For each dimension, we conducted two separate regression models: one using Poisson regression and one using Negative Binomial (NB) regression. This resulted in a total of four models—two for ME (Poisson and NB) and two for AR (Poisson and NB). Poisson and NB models, which are used for non-negative integers, were applied to model ME and AR. The difference between Poisson and NB is that Poisson assumes the mean and variance of the data are equal, while NB accounts for overdispersion by introducing a dispersion parameter that allows the variance to exceed the mean.

In both models, positive coefficients indicate an increasing trend in message fatigue associated with the predictor variable, while negative coefficients indicate a decreasing trend. For categorical variables, coefficients represent differences compared to the reference group. For example, for the sex variable, female is the reference group, so a positive coefficient for male indicates higher message fatigue compared to females, while a negative coefficient indicates lower message fatigue compared to females.

To identify the most suitable model, supervised backward selection with a $\chi^2$ test at a 0.05 significance level was employed. The Akaike Information Criterion (AIC) determined the final model based on the lowest value. If the 95% confidence interval of the coefficients estimated in Poisson and NB regressions does not include zero, the effect is considered statistically significant. The MASS (version 7.3.61) and car packages (version 3.1.3) were utilized, and all model selections and tests were conducted in R version 4.2.2 [32–34].

## Results

A total of 818 individuals started and completed the survey, yielding a 100% completion rate. Participants under the age of 18 (N = 4) were excluded, resulting in a final sample of 814 individuals.

Among the participants, females (N = 538) outnumbered males (N = 276). Most respondents had an education level higher than high school (N = 690), while a smaller portion had an education level equal to or below high school (N = 124). Additionally, a larger group reported no job experience in dengue control (N = 696) compared to those with relevant job experience (N = 118). Regarding the optimism bias variable, the majority of respondents were classified as realistic (N = 541), followed by those in the optimism (N = 209) and pessimism groups (N = 64) (Table 1). The age range of participants was from 18 to 88 years, with an average age of 46.2 years.

### High dengue risk (i.e., Southern region)

There were 466 participants in the high dengue risk region, aged 18–83, with an average age of 49.4 years. More participants were female (N = 310), had a higher education level than high school (N = 365), and lacked job experience in dengue control (N = 384) compared to males (N = 156), those with an education level equal to or below high school (N = 101), and those with job experience in dengue control (N = 82) (Table 1). The optimism bias variable indicated that the largest proportion of participants were in the realistic group (N = 308), followed by the optimism (N = 121) and pessimism groups (N = 37) (Table 1).

In terms of perceived risks, 83% and 58% of participants in the high-risk region rated perceived prevalence and perceived severity above 5 on an 11-point Likert scale (0 = Not serious at all, 10 = Extremely serious), respectively (Fig 1). For questions evaluating optimistic bias, over 50% of participants in both questions scored above 5 on the same scale (Fig 1).

For message fatigue, measured on a 7-point Likert scale ranging from 1 (strongly disagree) to 7 (strongly agree), the proportion of participants selecting responses between 4 and 7 (neutral to strongly agree) in the ME dimension was 64%, 45%, 52%, 49%, and 61% for questions ME1, ME2, ME3, ME4, and ME5, respectively (Fig 2). For the AR dimension, the proportions were 40%, 36%, 49%, and 45% for questions AR1, AR2, AR3, and AR4, respectively (Fig 2).

The adjusted GVIF values for each independent variable were less than two in the high-risk area, indicating no multicollinearity issues (S4 Table). For the ME dimension in the southern region, the negative binomial model (AIC = 3063.30) was preferred over the Poisson model (AIC = 3257.76) for evaluating associations. However, no specific significant predictors were identified for feelings of boredom and annoyance associated with dengue messages in southern Taiwan (Table 2).

**Table 1. Distribution of participants by sex, education, job experience, and optimism bias in different dengue risk areas in Taiwan (N = 814).**

| Rick area | | High* | | Low** | | Total | |
|---|---|---|---|---|---|---|---|
| **Variables** | **Categories** | **N** | **%** | **N** | **%** | **N** | **%** |
| Sex | Female | 310 | 67 | 228 | 66 | 538 | 66 |
| | Male | 156 | 33 | 120 | 34 | 276 | 34 |
| Education | Higher than high school | 365 | 78 | 325 | 93 | 690 | 85 |
| | Equal to or below high school | 101 | 22 | 23 | 7 | 124 | 15 |
| Job experience in dengue control | No | 384 | 82 | 312 | 90 | 696 | 86 |
| | Yes | 82 | 18 | 36 | 10 | 118 | 14 |
| Optimism bias | Realistic | 308 | 66 | 233 | 67 | 541 | 66 |
| | Optimism | 121 | 26 | 88 | 25 | 209 | 26 |
| | Pessimism | 37 | 8 | 27 | 8 | 64 | 8 |
| Total | | 466 | 100 | 348 | 100 | 814 | 100 |

*Southern region.

**Northern, Central, Eastern, and Outlying island regions.

Given the lower AIC, the negative binomial model (AIC = 2891.50) was chosen over the Poisson model (AIC = 3112.21) to evaluate associations related to message fatigue in the AR of the southern region. The positive coefficient (0.10, 95% CI = 0.01–0.19) indicated that individuals with an education level equal to or below high school reported significantly higher levels (p = 0.04) of boredom and annoyance compared to those with a higher education level in southern Taiwan (Table 2).

### Low dengue risk (i.e., Northern, Central, Eastern, and Outlying island regions)

There were 348 participants in the low dengue risk region. In this region, more participants were female (N = 228), had higher education levels than high school (N = 352), and lacked job experience in dengue control (N = 312) compared to males (N = 120), those with education levels equal to or below high school (N = 23), and those with job experience in dengue control (N = 36) (Table 1). The optimism bias variable indicated that the majority of participants fell into the realistic group (N = 233), followed by the optimism group (N = 88) and the pessimism group (N = 27) (Table 1). Ages ranged from 18 to 88 years, with an average age of 41.6 years.

Regarding perceived risks, 75% of participants in the low-risk region rated perceived prevalence above 5 on an 11-point Likert scale, while 61% did the same for perceived severity (Fig 1). For questions assessing optimistic bias, fewer than 50% of participants scored above 5 on both items (Fig 1).

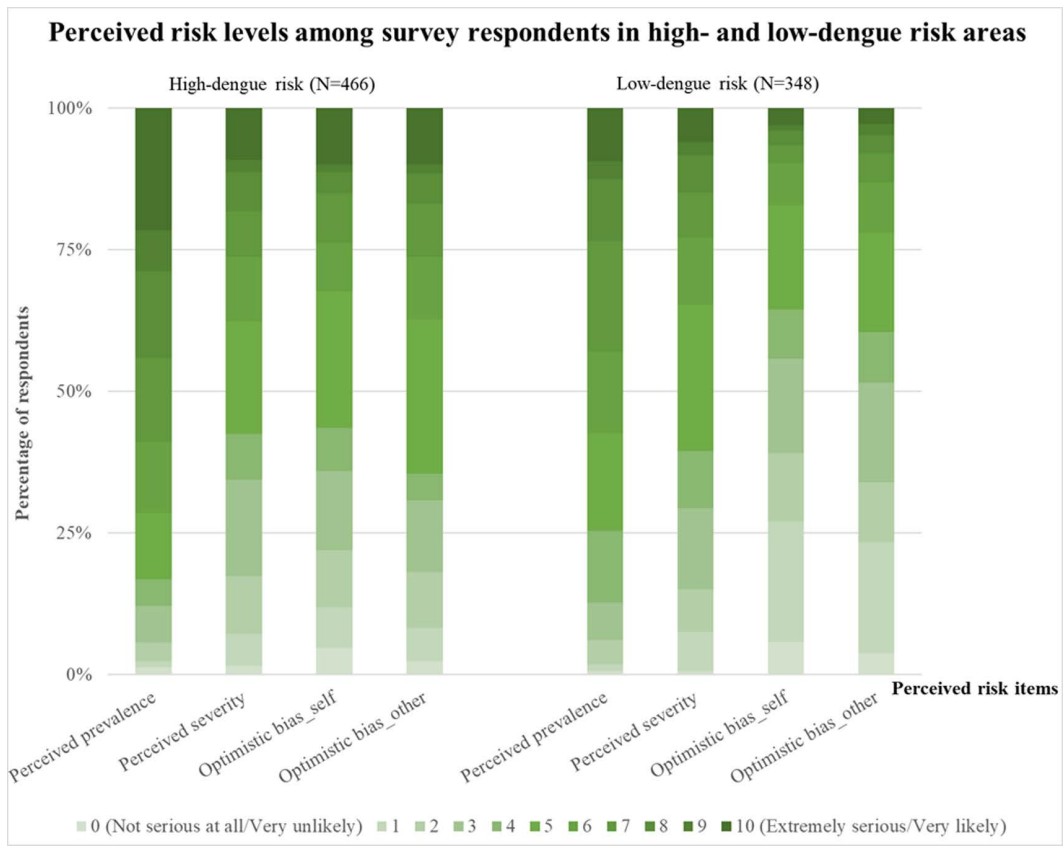

**Fig 1. Distribution of participants' perceived prevalence, perceived severity, and optimistic bias across different dengue risk regions in Taiwan (N = 814).** An 11-point Likert scale was used (0 = Not serious at all/Very unlikely, 10 = (Extremely serious/Very likely) (see Sections I and II in S1 Table). Optimistic bias was assessed using two questions: "How likely do you think it is that you will get a dengue infection?" (optimistic bias_self) and "How likely do you think it is that others around you will get a dengue infection?" (optimistic bias_other).

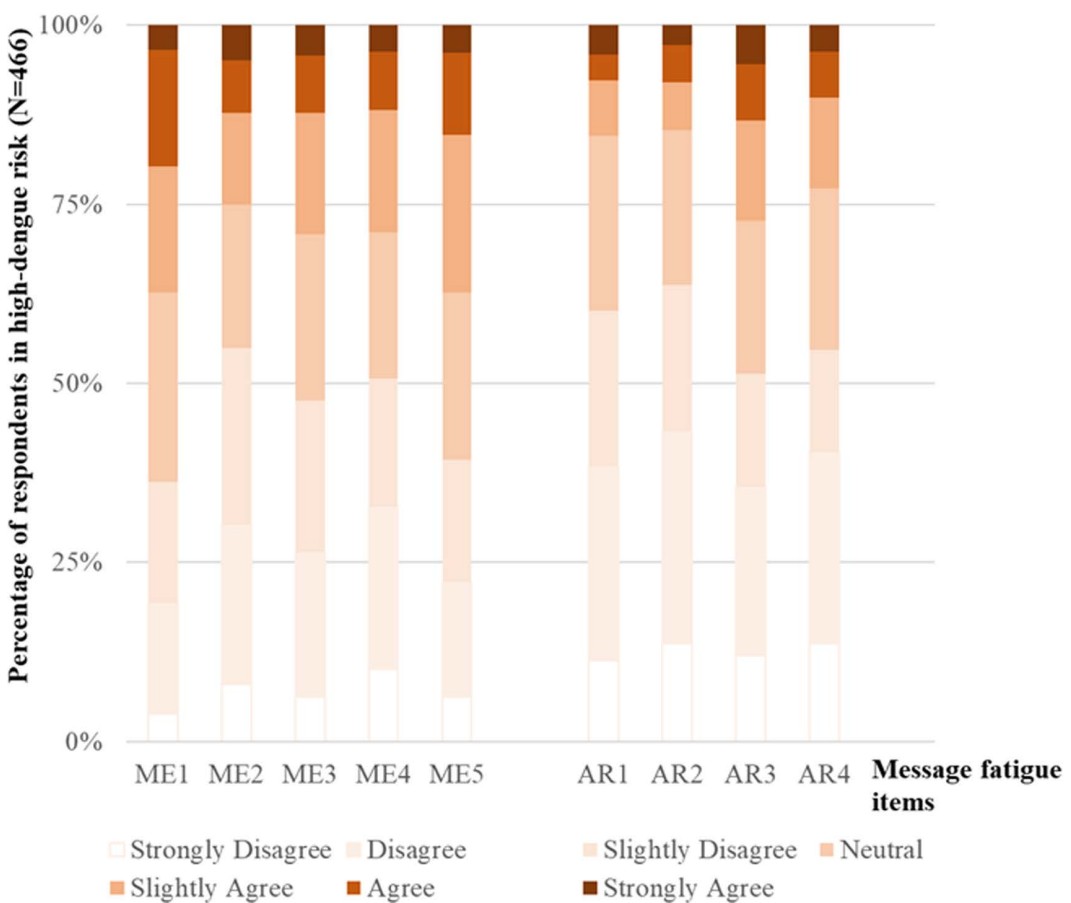

**Fig 2. Distribution of participants' responses to five questions in the message environment (ME) dimension and four questions in the audience responses (AR) dimension of message fatigue in a high-risk dengue region in Taiwan (N = 466).**

**Table 2. Negative binomial model for dengue message fatigue considering message environment and audience response in the high-risk area (N = 466).**

|  | Predictors | Reference group | Coeff[c] | SE[d] | 95% CI[e] | p |
|---|---|---|---|---|---|---|
| ME[a] | – |  | – | – | – | – |
| AR[b] | Education |  |  |  |  |  |
|  | Equal to or below high school | Higher than high school | 0.10 | 0.05 | 0.01–0.19 | 0.04 |

[a]Message Environment.

[b]Audience Response.

[c]Coefficient.

[d]Standard Error.

[e]Confidence Interval.

In the low-risk region, the proportion of participants who selected responses between 4 (neutral) and 7 (strongly agree) in the ME dimension was 52%, 34%, 40%, 39%, and 56% for ME1, ME2, ME3, ME4, and ME5, respectively (Fig 3). In the AR dimension, the corresponding proportions were 30%, 27%, 37%, and 39% for AR1, AR2, AR3, and AR4, respectively (Fig 3).

The adjusted GVIF values for each independent variable were less than two in the low dengue risk area, indicating no multicollinearity issues in the low dengue risk area (S4 Table). For ME in the low-risk regions, the negative binomial model was selected due to a lower AIC (AIC = 2173.6) compared to the Poisson model (AIC = 2260.1) for evaluating associations. Significant predictors included sex (p = 0.02), education (p = 0.01), and optimism bias (p = 0.02). Males (0.09, 95% CI = 0.02–0.17, p = 0.02) and individuals with education levels equal to or below high school (0.18, 95% CI = 0.04–0.32, p = 0.01) reported feeling more frustrated with an overabundance of dengue messages compared to females and people with higher education levels, significantly (Table 3). Additionally, the optimism group was more likely to feel frustrated by the large number of dengue messages than the realistic group (0.10, 95% CI = 0.02–0.18, p = 0.02). However, compared to the realistic group, the pessimism group did not report significantly more frustrated with the overabundance of dengue messages (0.12, 95% CI = -0.01–0.25, p = 0.07) (Table 3).

We opted for the negative binomial model (AIC = 2092.4) over the Poisson model (AIC = 2289.4) to assess associations related to message fatigue in the AR dimension due to the lower AIC. However, the application of the negative binomial model revealed no significant predictors for message fatigue in the AR dimension of the central region (Table 3).

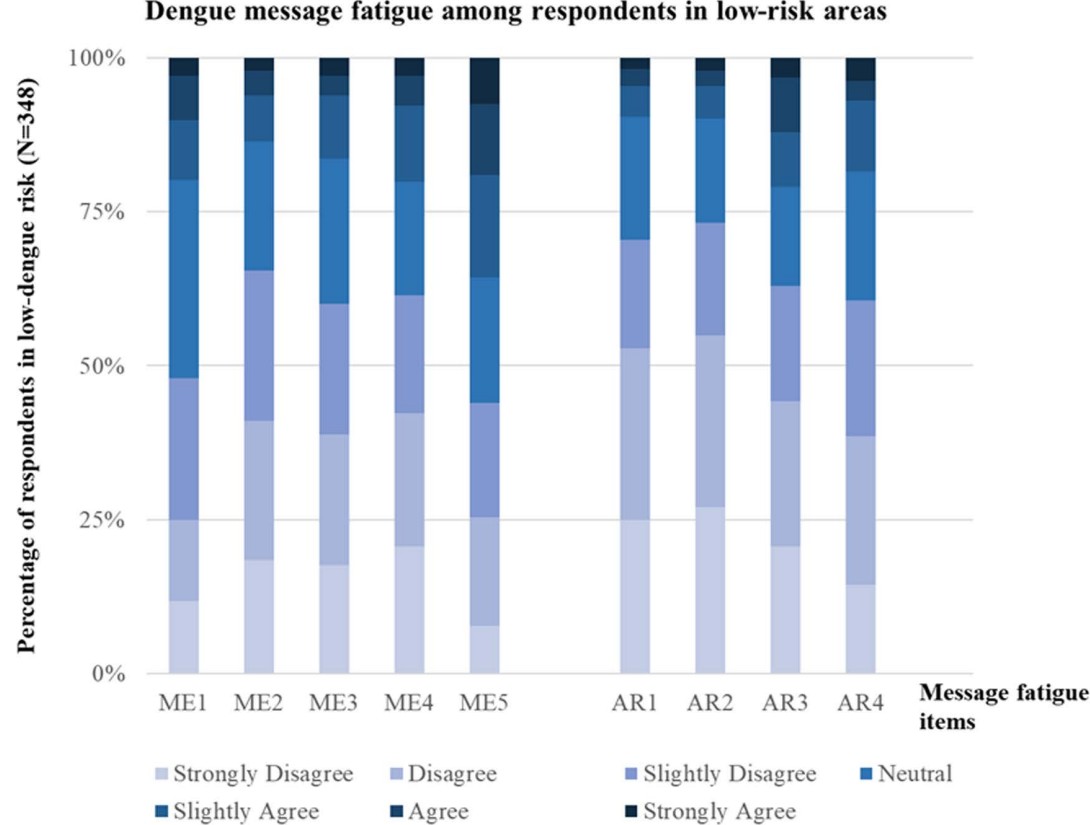

**Fig 3. Distribution of participants' responses to five questions in the message environment (ME) dimension and four questions in the audience responses (AR) dimension of message fatigue in a low-risk dengue region in Taiwan (N = 348).**

**Table 3. Negative binomial model for dengue message fatigue considering message environment and audience response in low-risk area (N = 384).**

|  | Predictors | Reference group | Coeff[c] | SE[d] | 95% CI[e] | p |
|---|---|---|---|---|---|---|
| ME[a] | Sex |  |  |  |  |  |
|  | Male | Female | 0.09 | 0.04 | 0.02–0.17 | 0.02 |
|  | Education |  |  |  |  |  |
|  | Equal to or below high school | Higher than high school | 0.18 | 0.07 | 0.04–0.32 | 0.01 |
|  | Optimistic bias |  |  |  |  |  |
|  | Pessimism | Realistic | 0.12 | 0.07 | -0.01–0.25 | 0.07 |
|  | Optimism | Realistic | 0.10 | 0.04 | 0.02–0.18 | 0.02 |
| AR[b] |  |  |  |  |  |  |
|  | – | – | – | – | – |  |

[a] Message Environment.

[b] Audience Response.

[c] Coefficient.

[d] Standard Error.

[e] Confidence Interval.

## Discussion

This cross-sectional survey study assessed the two dimensions of message fatigue across regions with varying levels of dengue risk and analyzed them in relation to demographic factors and perceived risks. The findings indicated that predictors varied between ME and AR across regions, with significant predictors identified only for ME in low-risk regions and only for AR in high-risk regions. In the high dengue-risk region, individuals with an education level equal to or below high school exhibited higher message fatigue in the AR dimension compared to those with higher education levels. Conversely, no significant predictors were identified for the ME dimension. In the low dengue risk region, sex, education, and optimism bias emerged as significant predictors for ME, while no significant predictors were identified for the AR dimension. This suggests that, in the ME dimension, males and individuals with an education level equal to or below high school experienced higher levels of message fatigue. Additionally, those in the optimism group were more likely to experience dengue message fatigue compared to those in the realistic group in the low dengue risk region.

Participants in this study perceived a higher prevalence of dengue in the high-risk region compared to the low-risk regions (Fig 1). This finding aligns with the availability heuristic [35] and Protection Motivation Theory (PMT) [36,37]. The availability heuristic is a cognitive shortcut in which people estimate the frequency or probability of an event based on how easily they can recall examples [35]. More memorable or easily recalled events appear more frequent or probable, even if this is not statistically accurate [35]. This heuristic was illustrated in the context of foodborne disease (FBD), where individuals with prior FBD experience relied on their memories rather than adjusting their risk estimates based on case numbers or external sources [38]. Even those who did not personally contract FBD but knew someone affected also exhibited higher risk perception [38]. Similarly, the availability heuristic may explain why individuals in high-risk dengue regions perceive a higher prevalence of dengue compared to those in low-risk regions. People in high-risk areas may perceive greater risk because they can easily recall past dengue cases, whereas those in low-risk areas may not due to a lack of firsthand experience. Another possible explanation is PMT, which elucidates how fear appeals (messages that warn about risks) influence attitude change and behavior [37]. PMT may clarify why people in high-risk areas perceive a higher disease prevalence by focusing on how they assess threats and their ability to take protective actions. Residents of high-risk areas often experience stronger threat and coping appraisals, leading to an increased perception of disease risk.

The findings of this study indicate that respondents in high-risk areas perceive a greater likelihood of dengue infection for both themselves and others compared to those in low-risk areas (Fig 1). However, despite this perception of risk, they do not regard dengue as more severe. This discrepancy can be explained by optimism bias, the habituation effect, and risk perception theory. Optimism bias refers to the tendency for individuals to believe they are less likely to experience health problems than others, even when objective risk factors suggest otherwise [39]. In this study, 56% and 65% of participants in high-risk areas who selected a score above 5 on an 11-point Likert scale perceived a high likelihood of dengue infection for themselves and others, respectively, indicating the presence of optimism bias. While these individuals acknowledge dengue's prevalence, they may assume that if they become infected, the illness will be mild. Even if they recognize the existence of severe cases, they may believe, "It won't happen to me." The habituation effect further elucidates why severity perception remains low despite high exposure. Repeated exposure to the same stimulus leads to desensitization, resulting in diminished emotional reactions over time [40]. Because dengue is a recurring event in high-risk areas, individuals may have adapted to its presence, reducing its perceived severity. Risk perception theory provides an additional explanation by examining how psychological and societal factors shape individuals' risk perceptions [41]. According to this theory, dread risk refers to the extent to which a risk is perceived as frightening, catastrophic, and uncontrollable. New or large-scale catastrophic risks tend to evoke high dread, whereas familiar and seemingly manageable risks generate lower levels of dread [41]. In the low-risk areas where dengue is less common, perceived severity remains similar to that in high-risk areas (Fig 1), possibly due to the higher dread risk associated with unfamiliar diseases. Because these individuals are not regularly exposed to dengue, they may view it as more unmanageable and potentially severe.

Higher levels of message fatigue in both the ME and AR dimensions are observed in high-risk regions, even when messages are delivered at the same frequency as in low-risk areas (Figs 2 and 3). This phenomenon can be explained through theories such as risk perception theory [41] and uses and gratifications theory (UGT) [42,43]. Risk perception theory posits that repeated exposure to a risk can diminish its perceived urgency over time, particularly when the risk becomes familiar and routine [41]. In this study, individuals in high-risk regions are continuously exposed to health messages, community discussions, and personal experiences related to dengue. This ongoing reinforcement could lead to risk habituation [40], wherein the threat becomes normalized, causing individuals to become desensitized to repeated warnings and perceive them as redundant, ultimately increasing message fatigue. In contrast, in low-risk regions, where disease outbreaks are infrequent, health messages are not reinforced as often through personal or social experiences. Consequently, individuals are less frequently exposed to the risk, making the same messages feel newer and more relevant, which helps maintain engagement and reduce fatigue. Another theory that may explain the higher message fatigue in high-risk regions compared to low-risk areas is UGT, which suggests that media consumption is goal-directed; audiences select content based on their expectations and the gratifications it provides [42,43]. People consume media to fulfill their social and psychological needs, which shape their engagement with different types of content [42,43]. Therefore, individuals in dengue high-risk regions may have different needs than those in low-risk areas, influencing their level of engagement and susceptibility to message fatigue. In high-risk regions, where individuals are already familiar with dengue prevention measures, there may have a greater demand for timely, detailed, and localized information to help navigate ongoing outbreaks. They are more likely to expect specific guidance on prevention and community measures rather than generic advice. Conversely, in low-risk regions, where individuals have less prior exposure to dengue, the same messages may still be perceived as valuable and worthy of attention.

Individuals with lower education levels experienced higher dengue message fatigue in the AR dimension in high-risk regions and in the ME dimension in low-risk regions in this study (Tables 2 and 3). Education levels have seldom been evaluated in relation to message fatigue in health contexts, except in a few areas such as anti-tobacco messaging [17]. The tendency for individuals with lower education levels to experience higher health message fatigue could be attributed to several factors, including mental effort, skills in handling information, and emotional toll. Regarding mental effort, individuals with lower education levels may have less prior knowledge or experience with health-related

topics. According to Cognitive Load Theory [44], the cognitive effort required to process, understand, and integrate new health information is greater for these individuals. This increased cognitive load may limit their ability to fully comprehend and engage with health messages, leading to mental exhaustion, confusion, and fatigue [44,45]. In terms of skills in handling information, individuals with higher education levels often possess better-developed information literacy skills, such as recognizing and utilizing information, enabling them to effectively evaluate and manage it [46]. This proficiency can mitigate feelings of being overwhelmed, whereas those with lower education levels might lack these skills, making them more susceptible to information overload. Additionally, the psychological impact of prolonged exposure to health messages can lead to burnout. Individuals with lower education levels may not have the coping mechanisms [47,48] or support systems [47,49] necessary to process ongoing health communications effectively. Therefore, they may experience this fatigue more acutely. To mitigate their fatigue, simple [44], relevant [50], and engaging [51] dengue messages should be considered.

In addition to individuals with lower education levels, males were identified as experiencing higher dengue message fatigue in the ME dimension in low dengue-risk regions compared to females. This finding is noteworthy because most studies report higher message fatigue in females [18,52,53], with fewer instances of fatigue observed in males [17]. In this study, males exhibited a greater level of message fatigue, which may be due to their lower likelihood of actively seeking health information in Taiwan [54,55]. Consequently, when males are exposed to health messages, they may find them more overwhelming, leading to quicker fatigue. Another possible explanation is that males may not engage with health messages on an emotional level as deeply as females do [56,57]. This lack of engagement could lower their threshold for perceiving messages as repetitive or unimportant, thereby increasing fatigue. Additionally, males may be more resistant to certain types of health messaging, particularly those that challenge their independence or self-reliance. For instance, messages that encourage using mosquito repellents, wearing long sleeves, or utilizing bed nets might be perceived as an imposition on their usual routines or as suggesting that they cannot protect themselves without these aids. This resistance can exacerbate fatigue, as they may view these messages as unnecessary.

According to this study, individuals in the optimism group were more likely to experience dengue message fatigue in the ME dimension than those in the realistic group in low-risk regions. Members of the optimism group tend to believe they are less likely to be affected by negative outcomes, such as dengue infection [58,59]. Therefore, even in areas with high prevalence, they may think, "It won't happen to me," which diminishes their perception of personal risk. This optimism bias could contribute to message fatigue, as it leads individuals to perceive health messages as irrelevant or unnecessary. Such perceptions reduce their engagement with the messages and increase their resistance to the content, resulting in fatigue. Furthermore, optimistic individuals may believe their actions are sufficient for protection, causing them to disregard additional advice or warnings. This overconfidence diminishes their receptivity to ongoing messages, leading to fatigue [60]. For this group, messages framed around potential losses could be more impactful, as they might help individuals take the situation more seriously and avoid complacency regarding their health and safety.

Poisson and NB models were applied in this analysis. While ordinal logistic regression is a common approach for analyzing Likert scale data, it assumes proportional odds across categories, which may not always hold in practice [61,62]. Additionally, it does not explicitly account for overdispersion. In contrast, NB models are specifically designed to handle overdispersion and are well-suited for modeling discrete, bounded, and count-like data. Poisson and NB models also focus on identifying trends and relationships without relying on the proportional odds assumption, making them more flexible for analyzing Likert responses in our context. For these reasons, Poisson and NB models were employed. However, caution is warranted when interpreting the results. Coefficients from these models should be understood as directional trends (e.g., higher values of predictor X are associated with increased message fatigue). If coefficients are described as changes in the expected count (e.g., males experience 1.5 times higher message fatigue than females), the interpretation may not always translate well to Likert scales.

## Conclusion

This study highlights how message fatigue in dengue prevention messaging varies across demographic groups and risk regions, emphasizing the need for tailored public health communication. In high-risk areas, individuals with lower education levels reported greater message fatigue in the AR dimension, indicating stronger feelings of exhaustion and tedium from repeated messaging. In low-risk areas, males, individuals with lower education levels, and those exhibiting an optimistic bias were more susceptible to message fatigue in the ME dimension, perceiving dengue messaging as excessive and redundant. These findings suggest that a uniform messaging approach may not be effective across different regions and populations, and communication strategies should be adapted accordingly to maintain engagement and effectiveness.

This study also uncovers several novel insights regarding risk perception and message fatigue. First, while public health messages are delivered uniformly across Taiwan, individuals in high-risk regions experienced greater message fatigue, suggesting habituation and desensitization rather than heightened engagement. Second, the relationship between education level and message fatigue varied by risk level—lower education levels were associated with AR fatigue in high-risk areas and ME fatigue in low-risk areas, indicating that both cognitive overload and information redundancy contribute to disengagement. Third, men in low-risk regions exhibited higher message fatigue than women, challenging prior research and suggesting that low engagement with health messaging may amplify frustration among men. Fourth, optimistic bias was associated with increased message fatigue, implying that individuals who perceive low personal risk are more resistant to health messages, making them more prone to fatigue. Lastly, despite higher perceived prevalence, individuals in high-risk areas did not perceive dengue as more severe than those in low-risk areas, likely due to risk habituation or optimistic bias.

These findings suggest important implications for public health messaging. In high-risk areas, localized and specific messages may be more effective than repetitive warnings, which can lead to desensitization. In low-risk areas, messages should focus on engagement and relevance, particularly for men and individuals with optimistic bias, to counteract perceptions of irrelevance. Moreover, simplified messaging strategies may help reduce cognitive overload and frustration among lower-educated populations. While this study did not measure behavioral responses, previous research suggests that message fatigue can contribute to disengagement from health messages, potentially influencing preventive behaviors.

Building on this study's strengths in identifying demographic and perception-based predictors of message fatigue, future research should examine the relationship between message fatigue and psychological reactance, as well as its potential impact on disengagement from preventive behaviors. To enhance generalizability, future studies should employ probability sampling methods, such as random or stratified sampling, to ensure a more representative population. Additionally, longitudinal and experimental studies should be conducted to establish causal relationships between message fatigue and behavioral responses. Implementation studies should assess how different communication strategies can be applied in real-world public health settings to reduce message fatigue and improve the reception of dengue prevention messages. Future research should explore whether modifying message frequency, format, or delivery platforms can sustain engagement without increasing fatigue.

By refining dengue messaging strategies based on patterns of risk perception and message fatigue, public health authorities can develop more effective, audience-centered communication approaches. However, without implementation studies to assess their effectiveness, it remains unclear whether such approaches will ultimately improve the reception and impact of dengue prevention messages.

## Limitations

While this study provides insights into message fatigue and its predictors across different dengue risk regions, several limitations should be considered. First, although this study focused on message fatigue as a factor influencing preventive

behaviors, the boomerang effect may also play a role, as individuals who perceive health messages as a threat to their freedom may resist them. Second, since participation was voluntary, selection bias may be present, particularly in gender and education level distribution. Compared to Taiwan's general population (51% female, 49% male; 42% with higher education) [63], our sample had a higher proportion of females (66%) and a greater percentage of highly educated individuals (85% with higher education). Therefore, caution is needed when applying these findings at a population-wide level. Third, the online questionnaire format may have led to misinterpretation of questions, potentially affecting data accuracy, though pretesting was conducted to minimize this risk. Fourth, recall bias may have influenced responses, as the survey was conducted at the end of the dengue season. In high-risk areas, prolonged exposure to outbreaks and messaging may have led to higher perceived risk and message fatigue, while in low-risk areas, participants may have underestimated their risk perception and message exposure. Lastly, this study identifies associations between message fatigue and various factors but does not establish causality.

## Supporting information

**S1 Table. Questionnaire.**
(PDF)

**S2 Table. Dimensions and questions of message fatigue.** Questions adapted from So et al. [13].
(PDF)

**S3 Table. Dataset (N = 814).**
(XLSX)

**S4 Table. Multicollinearity assessments of independent variables in each risk area.**
(PDF)

**S5 Table. Variables, codes, and categories in the dataset.**
(PDF)

**S6 Table. Cross check list [64].**
(PDF)

## Author contributions

**Conceptualization:** Chia-Hsien Lin, Yen-Jung Chang, Hung-Yi Lu.

**Data curation:** Chia-Hsien Lin.

**Formal analysis:** Chia-Hsien Lin.

**Investigation:** Chia-Hsien Lin, Hung-Yi Lu.

**Methodology:** Chia-Hsien Lin, Yen-Jung Chang.

**Project administration:** Chia-Hsien Lin.

**Software:** Chia-Hsien Lin.

**Validation:** Chia-Hsien Lin.

**Visualization:** Chia-Hsien Lin.

**Writing – original draft:** Chia-Hsien Lin.

**Writing – review & editing:** Yen-Jung Chang, Hung-Yi Lu.

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
