## [Decision Letter · Decision Letter 0]

3 Jan 2025

PNTD-D-24-01722

Different predictors of message fatigue across diverse risk areas for dengue prevention in Taiwan

Dear Dr. Lin,

Thank you for submitting your manuscript to PLOS Neglected Tropical Diseases. After careful consideration, we feel that it has merit but does not fully meet PLOS Neglected Tropical Diseases's publication criteria as it currently stands. Therefore, we invite you to submit a revised version of the manuscript that addresses the points raised during the review process.

Please submit your revised manuscript within 60 days Mar 04 2025 11:59PM. If you will need more time than this to complete your revisions, please reply to this message or contact the journal office at plosntds@plos.org. Please include the following items when submitting your revised manuscript:

We look forward to receiving your revised manuscript.

Kind regards,

Philip Mshelbwala

Academic Editor

Qu Cheng

Section Editor

Shaden Kamhawi

co-Editor-in-Chief

Paul Brindley

co-Editor-in-Chief

**Journal Requirements:**

At this stage, the following Authors/Authors require contributions: Chia-Hsien Lin. Please ensure that the full contributions of each author are acknowledged in the "Add/Edit/Remove Authors" section of our submission form.

**Reviewers' Comments:**

Reviewer's Responses to Questions

**Key Review Criteria Required for Acceptance?**

**Methods**

-Are the objectives of the study clearly articulated with a clear testable hypothesis stated?

-Is the study design appropriate to address the stated objectives?

-Is the population clearly described and appropriate for the hypothesis being tested?

-Is the sample size sufficient to ensure adequate power to address the hypothesis being tested?

-Were correct statistical analysis used to support conclusions?

-Are there concerns about ethical or regulatory requirements being met?

Reviewer #1: Major comments:

Materials and Methods:

1. Was the study conducted and reported in compliance with the Consensus -Based Checklist for Reporting of Survey Studies (CROSS) guidelines? [1]. This checklist should be completed and added to the Supplementary Information as survey data was used.

2. Lines 226-231 - Statistical analysis: The statistical analysis methods section should be expanded to improve interpretability of the results.

a. Were any of the independent variables correlated before including the variables into the models?

b. How were the ME and AR per participant data calculated in the S2 Data Supplementary File? Is the overall value calculated using number rankings for the six question for ME and five questions for AR for the five-point Likert scales? Does that mean that a higher score for ME and AR based on the questions in the S1 Table represented feeling higher levels of being overwhelmed and frustrated, and exhaustion and tedium, respectively?

This information isn’t explicitly mentioned in the manuscript and would help to improve interpretability. Please add how the values in S2 Data were calculated, particularly for ME and AR, in the statistical analysis methods section.

c. Likert scale data is generally ordinal, not continuous. Please expand further and justify the use of Poisson and negative binomial models in the methods section [2].

d. Lines 226-227: The authors stated that “The dimensions of ME and AR were assessed separately, as the questions described different feelings.”

Explicitly state whether two separate models were conducted for ME and AR for both Poisson and negative binomial models. This is currently unclear in the methods but is a clearer in the results.

e. Was optimism vs pessimism analysed in the models as an independent variable? In Table 2, the final predictors were pessimism vs realistic and optimism vs realistic. Please clearly state in the methods section what comparisons were made in the statistical analysis and what the reference group for each categorical variable is.

f. The estimated coefficients were exponentiated and represented in Tables 1 to 3. This should be mentioned this in the methods section of the manuscript.

g. Line 231: Reference the R packages and the versions of the R packages used to conduct the statistical analyses.

References:

1. Sharma, A.; Minh Duc, N.T.; Luu Lam Thang, T.; Nam, N.H.; Ng, S.J.; Abbas, K.S.; Huy, N.T.; Marušić, A.; Paul, C.L.; Kwok, J.; et al. A Consensus-Based Checklist for Reporting of Survey Studies (CROSS). J Gen Intern Med 2021, 36, 3179-3187, doi:10.1007/s11606-021-06737-1.

2. Sullivan, G.M.; Artino, A.R., Jr. Analyzing and interpreting data from likert-type scales. J Grad Med Educ 2013, 5, 541-542, doi:10.4300/jgme-5-4-18.

Reviewer #2: The manuscript clearly presents the objectives of the research and the study design is appropriate in addressing the researchers' objectives. The authors gave a clear description of the study population. The statistical analysis adopted were relevant and their applications were clearly explained. Relevant ethical approval was obtained, meeting regulatory requirements for a standard scientific research.

Reviewer #3: Major

1. The classification of high, medium, and low risk regions is not appropriate in this article. Define southern region as a high-risk region is fine because both vectors are coexisted. However, the definition of medium and low risk regions is questionable because both regions have the same single vector. Even though the classification was based on the cumulative case number in the past, the three regions are hard to be separated equally using risk levels. Accurate definition of risk levels is important in this paper because the author want to improve the effect of public health messages in regions at various risk levels.

2. The voluntary participants might introduce selection bias. What’s the age/gender/education level distribution of your study subjects? Can it comparable to the general population?

3. According to the questions about either ME and AR, they are more like the feeling about the messages. From the questionnaire, it does not associate with any action changes. People who has message fatigue of dengue control does not mean they don’t want to do anything to control the disease.

4. Regarding the “job experience (relevance to dengue control)” variable, what the reason to include this parameter? I thought you should include simply “job” information of each participant.

5. How do you treat the dependent variable in your model? Both ME and AR have multiple questions, how do you transform them in to your outcome (Y)?

Minor:

1. The demographic characteristic information of the participant can be presented as a table.

**Results**

-Does the analysis presented match the analysis plan?

-Are the results clearly and completely presented?

-Are the figures (Tables, Images) of sufficient quality for clarity?

Reviewer #1: Major comments:

Results:

1. Lines 259-262: The results of the survey for perceived prevalence and severity have been reported descriptively for ranges and averages, and the responses of the 801 participants has been provided in the Supplementary Information.

For the Likert scale data, participants select only one option per question, and these options are ordinal rather than continuous. Reporting averages for this data may be misleading as the intervals between options are not necessarily equal. A mean response of 6.5, 5.0, 6.9 etc., does not accurately reflect the distribution of the responses as it isn’t a continuous scale. It would be more informative to create Likert plots or represent the proportion of responses in a table for each risk area. This would allow the readers to better understand the overall distribution of message fatigue and perceived risks prior to the statistical analysis and provide a stronger descriptive representation.

Please produce Likert plots or tables for each risk area to show the proportion of participants selecting each possible answer and discuss appropriately throughout the manuscript.

Reviewer #2: There is alignment between the analysis presented and the analysis plan. All results were well presented with clear and adequate number of tables.

Reviewer #3: Overall, the article contained two major issues. 1. Sampling scheme and small sample size make it difficult to infer to general population. 2.No clear definition of medium and low risk levels. The three models gave you different outcomes. Any novelty or insight can be inferred from those outcomes?

If you can have the information about action change of dengue control regrading message fatigue, this article will be more valuable.

**Conclusions**

-Are the conclusions supported by the data presented?

-Are the limitations of analysis clearly described?

-Do the authors discuss how these data can be helpful to advance our understanding of the topic under study?

-Is public health relevance addressed?

Reviewer #1: (No Response)

Reviewer #2: The conclusions drawn from the study are in line with the data presented by the authors and the limitations of the study were stated with sincerity. The expression of the public health relevance of the research was concise and the authors discussed how the data obtained in their study could advance understanding of the topic.

Reviewer #3: Lastly, how do you apply the finding in the article to modify dengue control community education in Taiwan?

**Editorial and Data Presentation Modifications?**

Reviewer #1: Minor comments:

Materials and methods:

1. Lines 178-185 – Participants and procedure:

a. Lines 178-179: As this study was part of a larger project, has any of the data from the online survey been published previously and is the layout of the survey available? To ensure transparency, please cite the relevant studies and add the survey to the Supplementary Information.

2. Lines 213-215: “The score of the first question was subtracted from that of the second to categorize optimistic bias into three groups: realistic (score zero), optimistic (positive score), and pessimistic (negative score).”

Has this method been implemented by other studies for optimistic bias? Please justify the use of this method and add interpretations of realistic, optimistic and pessimistic to aid easier comprehension of the terminology in the context of this study.

Results:

1. Lines 246-250: Please report the completion rate of the survey. How many participants started the survey but didn’t answer all questions and how many answered all questions?

2. Line 318: The low dengue risk results section appears to be missing an initial sentence stating the total number of participants in that area, i.e. “The region of low dengue risk had 116 participants.”. Please add this sentence to be cohesive with the other two risk area sub-headings.

3. Lines 328-329: The authors stated that “For ME in the central region, due to a lower AIC, the negative binomial model was chosen (AIC = 735.75) instead of the Poisson model (AIC = 639.12) for evaluating the associations.” But the Poisson model had a lower AIC this time? Please amend appropriately.

4. Table 1, Table 2 and Table 3:

a. As the manuscript mentions “significantly higher levels” (line 286), report the p-values in the negative binomial model tables and text for all statistical results.

b. The abbreviated words in Tables 1 to 3 should be written in full in the footnotes for Est, SE, Exp(EST) and CI.

Discussion:

1. Lines 351-354: The manuscript states: “Combining ME and AR, in the high dengue-risk region, people with an education level equal to or below high school had higher message fatigue compared to those with a higher education level (above high school).”

If the ME and AR models were conducted separately, then this is the result for only the AR dimension/component of message fatigue and not both dimensions. Please reword this to be specific on which dimension is being discussed as it is stated that “no significant predictors were identified for the ME dimension” (line 350) so the dimensions shouldn’t be combined.

Please do the same, where appropriate, for the entire discussion.

2. Lines 355 to 357: Following on from above, please be specific on which message fatigue dimension the statistical analysis results refer to in the discussion. The dimensions were analysed separately as they refer to different, specific feelings of message fatigue.

For instance, the sentence could include “This indicates that people in the optimism group were more likely to experience dengue-message fatigue regarding the ME dimension compared to those in the realistic group.”

3. Lines 381-384: Remove the “are” from the end of the sentence in Line 384.

4. Lines 389-395: To help corroborate these sentences, it would be beneficial to state how dengue prevention messages were framed or delivered to the public in Taiwan prior to and during the availability of this survey.

5. Lines 434-446: For the limitations, the authors state that using a convenience sampling method might not accurately reflect the entire population and that questions may have been misinterpreted.

Please address any other biases that might exist in this survey data (i.e., not including eastern Taiwan, non-response and recall bias) and how they relate to this study.

6. Lines 434-466: Are there any future research directions from this study that could be mentioned such as implementation studies? This would help to link back to the conclusions/significance mentioned in the abstract and introduction.

7. Lines 432-434: The conclusion to the discussion appears to be at the start of the limitations paragraph and the discussion ends abruptly. To ensure that the conclusion is clearly and appropriately highlighted, please relocate these sentences to a new paragraph at the end of the discussion section and add any additional concluding information.

Supplementary Information:

1. S1 Table: In the manuscript, it is reported that six questions are used for ME and five questions are used for AR, but only five and four questions are reported in the table for S1 Table, respectively. Please amend this appropriately.

2. S3 Table: The reference group for each category in the models could be noted in this table.

3. Please make sure all Supplementary Information is referenced in the manuscript in the appropriate location.

Reviewer #2: Only minor modifications are required as indicated in the attached reviewed manuscript.

Reviewer #3: NA

**Summary and General Comments**

Reviewer #1: This manuscript identified distinct predictors of message fatigue across different dengue-risk areas in Taiwan, including high, medium and low risk areas using survey data. The survey count data for message environment (ME) and audience response (AR) was analysed using both Poisson and negative binomial models. All negative binomial models were selected as the final output reporting only significant factors in this manuscript.

I would like to acknowledge the considerable effort that has gone into this manuscript. Understanding different factors that play a role in message fatigue including demographic factors and perception risks is highly important. As the authors state in the introduction, the knowledge of message fatigue for dengue risk in Taiwan is sparse. These findings may aid in informing updated prevention messages with future targeted implementation research specific to high, medium and low risk areas to encourage public compliance.

As the statistical analysis forms the major component of this study, the analysis should be further technically expanded in the materials and methods section to understand how the Likert scale data was treated. Also, displaying the distribution of the Likert scale data for each risk area using Likert plots or tables would greatly aid in improving this study. The Likert plots or tables would allow readers to clearly understand the overall distribution of the participants responses for message fatigue and perceived risk prior to the statistical analysis. Expanding on the above will significantly strengthen this manuscript.

Reviewer #2: The research is a very unique one as there is paucity of peer reviewed articles on message fatigue regarding dengue fever. The topic is clear, the introduction is detailed and the entire manuscript is well structured and interesting to read. I strongly believe readers will also find it interesting and easy to comprehend. The research is of scientific and public health relevance as it provides insights for strategic and more community targeted preventive measures which would advance the goal of eliminating dengue fever in the study areas. I commend the efforts of the authors for this contribution to science and public health advancement

Reviewer #3: The manuscript entitled “Different predictors of message fatigue across diverse risk areas for dengue prevention in Taiwan” evaluated message fatigue of dengue education by EM and AR related questionnaire. Although the issue is worth to investigate, the study design and the content of the questionnaire make the results less insightful.

PLOS authors have the option to publish the peer review history of their article (what does this mean? ). If published, this will include your full peer review and any attached files.

**Do you want your identity to be public for this peer review?** For information about this choice, including consent withdrawal, please see our Privacy Policy .

Reviewer #1: No

Reviewer #2: No

Reviewer #3: No

**Figure resubmission:**
---

## [Decision Letter · Decision Letter 1]

31 Mar 2025

PNTD-D-24-01722R1Tailoring dengue health communication: Survey-based strategies to reduce message fatigue across risk areasPLOS Neglected Tropical Diseases Dear Dr. Lin, Thank you for submitting your manuscript to PLOS Neglected Tropical Diseases. After careful consideration, we feel that it has merit but does not fully meet PLOS Neglected Tropical Diseases's publication criteria as it currently stands. Therefore, we invite you to submit a revised version of the manuscript that addresses the points raised during the review process. Please submit your revised manuscript within 30 days Apr 30 2025 11:59PM. If you will need more time than this to complete your revisions, please reply to this message or contact the journal office at plosntds@plos.org. Please include the following items when submitting your revised manuscript:

* A rebuttal letter that responds to each point raised by the editor and reviewer(s). You should upload this letter as a separate file labeled 'Response to Reviewers '. This file does not need to include responses to any formatting updates and technical items listed in the 'Journal Requirements' section below.

 * A marked-up copy of your manuscript that highlights changes made to the original version. You should upload this as a separate file labeled 'Revised Manuscript with Track Changes '. * An unmarked version of your revised paper without tracked changes. You should upload this as a separate file labeled 'Manuscript '. If you would like to make changes to your financial disclosure, competing interests statement, or data availability statement, please make these updates within the submission form at the time of resubmission. Guidelines for resubmitting your figure files are available below the reviewer comments at the end of this letter. We look forward to receiving your revised manuscript. Kind regards, Philip P. MshelbwalaAcademic EditorPLOS Neglected Tropical Diseases Qu ChengSection EditorPLOS Neglected Tropical Diseases

Shaden Kamhawi

co-Editor-in-Chief

Paul Brindley

co-Editor-in-Chief

**Reviewers' comments:** Reviewer's Responses to Questions

**Key Review Criteria Required for Acceptance?**

**Methods:**

-Are the objectives of the study clearly articulated with a clear testable hypothesis stated?

-Is the study design appropriate to address the stated objectives?

-Is the population clearly described and appropriate for the hypothesis being tested?

-Is the sample size sufficient to ensure adequate power to address the hypothesis being tested?

-Were correct statistical analysis used to support conclusions?

-Are there concerns about ethical or regulatory requirements being met?

Reviewer #1: (No Response)

**Results:**

-Does the analysis presented match the analysis plan?

-Are the results clearly and completely presented?

-Are the figures (Tables, Images) of sufficient quality for clarity?

Reviewer #1: (No Response)

**Conclusions:**

-Are the conclusions supported by the data presented?

-Are the limitations of analysis clearly described?

-Do the authors discuss how these data can be helpful to advance our understanding of the topic under study?

-Is public health relevance addressed?

Reviewer #1: (No Response)

**Editorial and Data Presentation Modifications?**

Reviewer #1: Minor comments:

Lines 80-82: Consider rewording to this sentence to include “where” or may be a full stop after the word pronounced.

“In high-risk regions, message fatigue is more pronounced, where repetitive warnings may contribute to disengagement, suggesting a need for communication strategies that reduce redundancy and emphasize localized, actionable information.”

Lines 227-229: Is it five or four questions for AR dimensions? Was one of the five questions removed? Please amend appropriately.

“For the AR dimension, the number of questions was reduced from seven to five, representing exhaustion and tedium for the same reasons (Section VI 7–11 in S1 Table), with a Cronbach’s alpha of 0.88 for the four retained questions (S2 Table).”

Lines 577-578: Where is this information reported in the results?

“In this study, younger individuals in low dengue-risk regions experienced higher levels of dengue message fatigue in the ME dimension.”

Figure 1: While the results section mentions that an 11-point Likert scale was used (0 = strongly disagree, 10 = strongly agree) (Lines 373-374), to ensure that Figure 1 is self-explanatory, the scale numbers (0 to 10) should be complemented with corresponding descriptors (i.e., strongly disagree to strongly agree), similar to Figures 2 and 3.

Figures 1-3: To ensure clarity and self-explanatory design, Figures 1, 2 and 3 should include a short, relevant title along with a labelled y-axis and x-axis.

Suggestions:

Lines 577-605: Since only high and low risk areas were evaluated and age isn’t a significant factor anymore, please consider revising this paragraph to be more concise while retaining key information about the potential need to adjust how dengue risk is communicated to the public.

Lines 702-706: Consider rewording this sentence to include the below because public health authorities can develop different, audience-centred communication approaches using risk perception and message fatigue. However, without implementation studies to identify effectiveness, it remains unclear whether these approaches will improve the reception and impact of dengue prevention messages.

“By refining dengue messaging strategies based on risk perception and message fatigue patterns, public health authorities can aim to develop more effective, audience-centered communication approaches, which may improve the reception and impact of dengue prevention messages.”

**Summary and General Comments:**

Reviewer #1: Thank you to the authors for their immense effort and time taken to address all of the reviewer comments. Each question has been thoroughly addressed. The measurements of variables and statistical analysis methods sections have been expanded which have enhanced interpretability of the results. The inclusion of the Likert plots clearing indicate the responses of the participants and help bring clarity to the study.

The authors have made commendable improvements to the discussion by elaborating on the findings but maintaining conciseness within the discussion section, where possible, is advised. Overall, only a few minor changes and suggestions remain to further refine the manuscript.

PLOS authors have the option to publish the peer review history of their article (what does this mean? ). If published, this will include your full peer review and any attached files.

**Do you want your identity to be public for this peer review?** For information about this choice, including consent withdrawal, please see our Privacy Policy .

Reviewer #1: No

---

## [Decision Letter · Decision Letter 2]

14 May 2025

Dear Lin,

We are pleased to inform you that your manuscript 'Tailoring dengue health communication: Survey-based strategies to reduce message fatigue across risk areas' has been provisionally accepted for publication in PLOS Neglected Tropical Diseases.

Best regards,

Philip P. Mshelbwala

Academic Editor

Qu Cheng

Section Editor

Shaden Kamhawi

co-Editor-in-Chief

Paul Brindley

co-Editor-in-Chief

Reviewer's Responses to Questions

**Key Review Criteria Required for Acceptance?**

**Methods**

-Are the objectives of the study clearly articulated with a clear testable hypothesis stated?

-Is the study design appropriate to address the stated objectives?

-Is the population clearly described and appropriate for the hypothesis being tested?

-Is the sample size sufficient to ensure adequate power to address the hypothesis being tested?

-Were correct statistical analysis used to support conclusions?

-Are there concerns about ethical or regulatory requirements being met?

Reviewer #1: (No Response)

**Results**

-Does the analysis presented match the analysis plan?

-Are the results clearly and completely presented?

-Are the figures (Tables, Images) of sufficient quality for clarity?

Reviewer #1: (No Response)

**Conclusions**

-Are the conclusions supported by the data presented?

-Are the limitations of analysis clearly described?

-Do the authors discuss how these data can be helpful to advance our understanding of the topic under study?

-Is public health relevance addressed?

Reviewer #1: (No Response)

**Editorial and Data Presentation Modifications?**

Reviewer #1: (No Response)

**Summary and General Comments**

Reviewer #1: (No Response)

PLOS authors have the option to publish the peer review history of their article (what does this mean? ). If published, this will include your full peer review and any attached files.

**Do you want your identity to be public for this peer review?** For information about this choice, including consent withdrawal, please see our Privacy Policy .

Reviewer #1: No

---

## [Editor Report · Acceptance letter]

Dear Lin,

We are delighted to inform you that your manuscript, "Tailoring dengue health communication: Survey-based strategies to reduce message fatigue across risk areas," has been formally accepted for publication in PLOS Neglected Tropical Diseases.

Best regards,

Shaden Kamhawi

co-Editor-in-Chief

Paul Brindley

co-Editor-in-Chief
